# Comparison of Bitterness Intensity between Prednisolone and Quinine in a Human Sensory Test Indicated Individual Differences in Bitter-Taste Perception

**DOI:** 10.3390/pharmaceutics14112454

**Published:** 2022-11-14

**Authors:** Mengyan Deng, Noriko Hida, Taigi Yamazaki, Ryo Morishima, Yuka Kato, Yoshiaki Fujita, Akihiro Nakamura, Tsutomu Harada

**Affiliations:** 1Division of Pharmaceutics, Department of Pharmacology, Toxicology and Therapeutics, Graduate School of Pharmacy, Showa University, 1-5-8 Hatanodai, Shinagawa-ku, Tokyo 142-8555, Japan; 2Department of Clinical Pharmacy, Division of Clinical Research and Development, School of Pharmacy, Showa University, 6-11-11 Kita-karasuyama, Setagaya-ku, Tokyo 157-8577, Japan; 3Clinical Research Institute for Clinical Pharmacology and Therapeutics, Showa University Karasuyama Hospital, 6-11-11 Kita-karasuyama, Setagaya-ku, Tokyo 157-8577, Japan; 4Division of Pharmaceutics, Department of Pharmacology, Toxicology and Therapeutics, School of Pharmacy, Showa University, 1-5-8 Hatanodai, Shinagawa-ku, Tokyo 142-8555, Japan

**Keywords:** bitterness of drugs, bitterness sensitivity, medication adherence, clinical research, generalized Labeled Magnitude Scale (gLMS), taste receptors, TAS2Rs, polymorphisms

## Abstract

Prednisolone is a frequently prescribed steroid with a bitter, unpalatable taste that can result in treatment refusal. Oral suspensions or powder dosage forms are often prescribed, particularly to pediatric patients, as they improve swallowability and ease of dose adjustment. Consequently, the bitterness of prednisolone is more apparent in these dosage forms. Few studies have investigated prednisolone’s bitterness. Thus, in this study, 50 adults evaluated the bitterness of prednisolone using the generalized Labeled Magnitude Scale (gLMS), in comparison with quinine, a standard bitter substance. Overall, prednisolone-saturated solution demonstrated the same extent (mean gLMS score: 46.8) of bitterness as 1 mM quinine solution (mean gLMS score: 40.1). Additionally, large individual differences were observed in the perception of the bitterness of prednisolone and quinine. Perceived flavors of some drugs are reportedly associated with bitter-taste receptor (TAS2Rs) polymorphisms. Therefore, we investigated the relationship between subjects’ genetic polymorphisms of *TAS2R19*, *38*, and *46*, and their sensitivity to bitterness. Although a relationship between *TAS2R19* polymorphisms and the perception of quinine bitterness was observed, no significant relationship was found between the perceived bitterness of prednisolone and the investigated genes. Ultimately, the results show that despite individual differences among subjects, the cause of prednisolone’s strong bitterness is yet to be elucidated.

## 1. Introduction

Medication adherence is a critical healthcare issue, as poor adherence to therapy often results in negative health outcomes. Drug taste is an important factor that determines medication adherence [1,2], particularly in pediatric drug treatment. It has been reported that a drug having a particularly bitter taste can lead to a marked decrease in medication adherence [3]. Drugs can be formulated and encapsulated to minimize the bitterness of their ingredients. However, given that pediatric dosing is commonly determined based on body weight and the inability of pediatric patients to swallow capsules and tablets, liquid or other dispersed dosage forms of medication are preferred. Furthermore, solid pharmaceutical preparations intended for adults may be crushed or dissolved for administration to children, a process that further emphasizes the bitterness of the drugs [4].

Prednisolone is a drug that is widely used clinically for multiple disease states in patients of all ages. Some pediatric indications of prednisolone include treatment of chronic adrenal dysfunction, acute phase of Kawasaki disease, nephrotic syndrome, severe infectious disease, and malignant lymphoma. In Japan, powders are often used for treating pediatric patients as the dosage of powders can be easily adjusted; however, powder bitterness may lead to refusal. Two studies have explored the palatability of various oral dosage forms of prednisolone compared to that of other corticosteroids in pediatric patients aged 2 to 16 years using the facial hedonic scale, and their results corroborate the poor taste of this drug [5,6]. However, in these surveys, the bitterness of prednisolone was evaluated by administering oral tablets or liquid formulations containing flavored syrups; thus, the bitterness of prednisolone itself has not yet been fully investigated. Moreover, the bitterness of prednisolone has not yet been evaluated and compared with that of standard substances whose bitterness is known, such as quinine, to the best of our knowledge.

The objective of this study was to confirm the degree of bitterness of prednisolone. The secondary purpose was to determine whether there are differences in the sensitivity to the bitterness of this drug between individuals and to investigate potential causes of this. The implementation of a human sensory test was considered a suitable method for achieving the secondary purpose in particular, as mechanical taste sensors [7] are limited in terms of identifying individual differences in human taste perception and are thus not suitable for the purpose.

In humans, bitter taste is mediated by TAS2Rs, which are G-protein-coupled receptors on the taste bud cells of the tongue [8]. There are 25 subtypes of TAS2Rs [9]; each TAS2R recognizes multiple bitter substances, and one bitter substance may activate multiple TAS2Rs. The resulting diversity in substrate specificity allows a limited number of receptors to respond to a wide variety of bitter substances. Genetic polymorphisms have been identified in TAS2R genes, some of which are known to cause functional changes in bitter-taste sensitivity [10]. For example, variations in the *TAS2R38* gene affect the taste sensitivity to phenylthiocarbamide (PTC) and propylthiouracil (PROP).

In this study, bitterness intensities of prednisolone solutions were compared with those of quinine by sensory tests in adults using the generalized Labeled Magnitude Scale (gLMS) [11], a common scale for sensory tests. In addition, the bitterness of the powder dosage form frequently used for children in Japan was evaluated to elucidate the bitterness actually perceived by pediatric patients. Further, we investigated whether the bitterness of prednisolone is affected by genetic polymorphisms of *TAS2R38*, together with *TAS2R19* and *TAS2R46*, in which mutations have already been reported.

## 2. Materials and Methods

### 2.1. Subjects

This clinical study was conducted at the Showa University Clinical Research Institute for Clinical Pharmacology and Therapeutics and was approved by the Research Ethics Review Board of the Showa University (approval number: 326). The study was conducted with 50 participants who received explanations, consented, and confirmed their eligibility. The study was carried out from April to May 2021.

### 2.2. Eligibility

The inclusion and exclusion criteria are described in the following subsections.

#### 2.2.1. Inclusion Criteria

Consenting patients were eligible for this study if:Their age at the time of obtaining consent was between 20 to 55 years; andThey gave written consent to participate in this study.

#### 2.2.2. Exclusion Criteria

The following patients were ineligible for this study:Persons with a medical history that may affect the evaluation and safety of this study (drug abuse/dependence, alcohol abuse/dependence, gastrointestinal/cardiac/hepatic/renal/pulmonary/hematological disease, etc.); orPersons with a history of food allergies or drug hypersensitivity (including allergies); orThose who are currently smoking at the time of obtaining consent or those who have been quitting smoking for less than one year; orPersons with, or suspected of having an infectious disease requiring systemic or topical treatment; orThose who have had taste or smell abnormalities within 4 weeks of the study start date; orThose who are otherwise deemed ineligible at the discretion of the research doctor.

### 2.3. Stimuli

The five stimuli described in Table 1 were used for the sensory test.

Stimuli were prepared using quinine hydrochloride “Hoei” (Pfizer Japan Inc., Tokyo, Japan), Prednisolone for Biochemistry (product code: 165-11491; FUJIFILM Wako Pure Chemical Co., Ltd., Tokyo, Japan), purified water, and lactose. For the purposes of creating a “prednisolone-saturated solution,” 100 mg of prednisolone was weighed, mixed with 200 mL of purified water, agitated using a stirrer for 30 to 60 min, and subsequently filtered. The solution prepared by this method contained 293 µg/mL prednisolone (0.813 mM; the mean result of three HPLC quantification analyses).

The concentration of quinine was selected based on previous research. A concentration of 0.2 mM prednisolone was selected, as this is approximately 10 times weaker than that described in Pharmaceutical CODEX 1994: 1 g of prednisolone dissolved in 1300 mL of water (2.134 mM) [12]. The prednisolone-saturated solution was set as the higher-concentration stimulus. In this study, prednisolone powder was triturated 100 times with lactose to replicate the 1% formulation of prednisolone that is widely used for pediatric patients in clinical practice, and lactose was selected as the filler as it is the most common excipient used with prednisolone powder.

Liquid stimuli were placed into small cups and powder stimuli were put into small packages by a packaging machine.

### 2.4. Sensory Test

For the evaluation of the study, the gLMS, subject background information, and adverse events were recorded. For the gLMS, evaluation forms consisted of a 150 mm line from the bottom to the top of the page with a scale including “barely detectable” (2.1 mm; 1.4 units), “weak” (9 mm; 6 units), “moderate” (25.5 mm; 17 units), “strong” (52.05 mm; 34.7 units), “very strong” (78.75 mm; 52.5 units), and “strongest imaginable sensation of any kind” (150 mm; 100 units). The subjects did not receive any particular sensory panel training before enrolment.

The order of stimuli evaluation was randomized on each study date by designated staff that oversaw allocation. This was a single-blind study in which only the subjects did not know the test order. Thus, on the morning of the test, the investigator opened the envelope containing the test order designation, confirmed the order, and performed it.

The taste and aftertaste of each stimulus was evaluated by the gLMS. Unaided, subjects put each stimulus into their mouth, maintained stimuli in the mouth for 10 s, and then spit it out. Powder application was standardized to be placed in the middle of subjects’ tongues, and when spitting it out, subjects gargled for 1 or 2 s with water. Immediately after spitting out each stimulus, the taste was evaluated by the gLMS before gargling with sufficient water (about 200 mL). Five minutes after the first evaluation, the aftertaste of the stimulus was evaluated by the gLMS. At least 20 min was allowed to elapse between different stimuli, and a new test stimulus was not provided until it was confirmed that the taste of the previous stimulus did not remain.

### 2.5. DNA Extraction, PCR, and Sequencing

Oral mucosal cells were collected from the oral cavity of the subjects using cotton swabs. DNA was extracted from these cells using a MonoFas^®^ DNA Purification kit (Animos Corporation, Saitama, Japan) in accordance with the manufacturer’s handbook and centrifugation guidelines. Based on a previous report [13], genomic regions, including the coding sequences of *TAS2R19*, *TAS2R38*, and *TAS2R46*, were amplified by PCR, wherein corresponding sets of primers were used (Appendix A). The PCR products were directly sequenced using the BigDye^®^ Terminator v3.1/1.1 Cycle Sequencing kit (Applied Biosystems, Waltham, MA, USA) to determine the nucleotide sequences.

### 2.6. Statistical Analysis

Statistical analysis was performed using a JMP Pro 16.0.0 (SAS Institute Inc., Cary, NC, USA). The Kruskal–Wallis test was used to compare the gLMS results of the five stimuli, and the Wilcoxon rank-sum test was used to compare the results of two stimuli. The Wilcoxon rank-sum test was used to test for differences in the gLMS results by sex, and the Kruskal–Wallis test was used to test for differences in the gLMS results by gene. Fisher’s exact test was used to determine the relationship between genes and the gLMS, divided into three bitterness intensity categories. Statistical significance was set at *p* < 0.05.

## 3. Results

### 3.1. Subject Background

The age of the subjects who participated in the study ranged from 20 to 54 years (mean age: 28 years). There were twenty-five male subjects (seventeen in their twenties, six in their thirties, one in their forties, and one in their fifties) and twenty-five female subjects (eighteen in their twenties, six in their thirties, and one in their forties).

### 3.2. Bitterness Intensity of the Stimuli

The mean gLMS values for the bitterness intensity rated by the subjects, from smallest to largest, were 0.1 mM quinine (mean: 15.9, between “weak” and “moderate”), 0.2 mM prednisolone (mean: 24.5, between “moderate” and “strong”), prednisolone powder (mean: 34.1, between “moderate” and “strong”), 1 mM quinine (mean: 40.1, between “strong” and “very strong”), and prednisolone-saturated solution (mean: 46.8, between “strong” and “very strong”), as shown in Table 2. It was observed that bitterness scores for quinine and prednisolone in aqueous solutions increased with concentration. There was a statistically significant difference in the gLMS evaluation results between the prednisolone-saturated solution and powder, with the saturated solution having a stronger bitter taste (*p* = 0.0044).

A wide range of individual differences in bitterness rating was observed for each stimulus, shown in the maximum and minimum gLMS scores in Table 2, as well as in Figure 1. For example, the maximum and minimum values for prednisolone-saturated solution were 92.0 and 6.0, respectively, and 90.0 and 8.7, respectively, for 1 mM quinine. This was also the case for prednisolone powder. Despite having the lowest mean gLMS scores, the maximum and minimum values were 94.7 and 2.0, respectively, which were comparable to the results of the prednisolone-saturated solution.

The gLMS scores of the aftertastes were, in ascending order by mean value, 0.1 mM quinine (mean: 2.5, between “barely detectable” and “weak”), 0.2 mM prednisolone (mean: 5.3, between “barely detectable” and “weak”), prednisolone-saturated solution (mean: 9.5, between “weak” and “moderate”), prednisolone powder (mean: 9.6, between “weak” and “moderate”), and 1 mM quinine (mean: 12.9, between “weak” and “moderate”; as shown in Appendix A). Even after the subjects gargled with water and waited approximately 5 min on average, bitterness still lingered for some stimuli.

To examine whether there were differences in the perception of the bitterness of prednisolone and quinine in each subject, gLMS scores for the three stimuli (prednisolone solution, prednisolone powder, and quinine solution) were classified into three categories (“strong”: equal to or more than 34.7; “medium”: equal to or more than 6.0 and less than 34.7; and “weak”: less than 6.0). As shown in Figure 2, a total of nine patterns were observed in the bitterness susceptibility of the three stimuli: the pattern of rating the bitterness of both quinine and prednisolone as “strong” (patterns a–c); the pattern of rating only quinine as “strong” (pattern d); the pattern of rating prednisolone solution as “strong” while rating the intensity of the powder in various ways (patterns e–g); and the pattern where none of the samples were rated as “strong” (pattern h–i). The number of subjects who rated both quinine and prednisolone (either saturated solution or powder) as “strong” (patterns a–c) accounted for 42% of the total, while those who rated none of the samples as “strong” (patterns h–i) accounted for 14%. These results show that subjects who perceived the bitterness of quinine strongly did not necessarily perceive the bitterness of prednisolone strongly, and vice versa. Overall, there were more subjects who perceived both quinine and prednisolone as strongly bitter than those who perceived neither as strongly bitter.

### 3.3. Genes and gLMS Results

The relationship between the bitterness intensity of quinine and prednisolone rated by the gLMS and genotypes of *TAS2R19*, *TAS2R38*, and *TAS2R46*, were investigated. Firstly, no statistically significant differences were found in the gLMS scores of quinine and prednisolone between subjects with normal receptors and those with receptor gene variants. Secondly, when the gLMS was divided into three categories as previously described (Figure 2), a relationship was visible between the perceived bitterness of the quinine and the genetic variant *TAS2R19* (rs10772420) (*p* < 0.05) (Table 3). However, no relationship was observed between the perceived bitterness of quinine and variants in the other TAS2R genes, nor between the perceived bitterness of prednisolone and any of the gene variants investigated in this study (Appendix A).

### 3.4. Sex and gLMS Results

In relation to differences in the perception of bitterness by sex, females were more inclined to have higher mean gLMS scores for most of the stimuli. However, there were no statistically significant differences in the gLMS scores between males and females for any stimulus (Appendix A).

## 4. Discussion

In this study, the bitterness intensity of two different concentrations of prednisolone (0.2 mM and approximately 0.813 mM [mean of the saturated concentration]) was investigated using the gLMS. Our results show a high, concentration-dependent bitterness intensity of prednisolone, with an average score of 24.5 for the lower concentration and 46.8 for the higher concentration. In addition, in the present study, quinine, a common bitter drug used in sensory testing, showed an average gLMS score of 40.1, which is comparable to the previously reported value of ~50 at 1 mM [14]. The above results strongly support the conclusion that prednisolone exhibits a bitter taste comparable to quinine. However, the gLMS scores varied between each subject. Using a prednisolone-saturated solution as an example, the gLMS scores varied from a maximum of ninety-two to a minimum of six, demonstrating large individual differences in the perception of bitterness. In addition, there were patterns in whether a subject perceived quinine or prednisolone as strongly bitter or not, with a certain number of subjects falling into each pattern (as shown in Figure 2). These results suggest that the processes implicated in the perception of quinine and prednisolone bitterness do not necessarily stem from the same pathway. Although quinine is commonly used in sensory tests, the results show that it cannot be regarded as a standard indicator for the prediction of bitterness prior to taking other drugs in the clinical field.

The results of this study suggest the presence of prednisolone-specific taste receptors, although these may partially overlap with quinine-responsive receptors. In addition, individual differences in the genes encoding taste receptors may also result in a large difference in subjects’ perception. One example of a genotype that affects the degree of bitterness perception is the genetic polymorphism of *TAS2R38*. It has been reported that humans with the homozygous *AVI* variant of the *hTAS2R38* gene (*AVI*/*AVI* genotype) have significantly reduced taste sensitivity to the bitter substance PTC [15].

Thus, we investigated whether polymorphisms of the bitter-taste receptor genes are related to the variation in taste sensitivity observed in this study. In addition to *TAS2R38*, *TAS2R19*, which has been reported to affect bitterness sensitivity to quinine [16,17], and *TAS2R46*, which is known to have a null mutation on the 250th tryptophan in the amino acid sequence that is changed to a termination codon [13], were examined. While there was a relationship between quinine sensitivity and the *TAS2R19* genotype, corroborating the results of similar past studies, no association was found between prednisolone and variations in any of the *TAS2R* genes investigated. In addition to bitter receptors, other factors have also been reported to determine bitterness sensitivity, such as sex [18], age [19], and lifestyle [20]. In relation to sex differences, it has been reported that the number of taste receptors on fungiform papillae was larger in females, and that this number was positively correlated with the perception of the bitterness of PROP, a widely used bitter substance in research [21]. Although we found that the mean gLMS values tended to be higher in females, there were no statistically significant differences between males and females. Unfortunately, the results of this study did not reveal individual characteristics (genotypes, sex, etc.) related to the perception of the bitter taste of prednisolone. Nevertheless, obtaining such information may lead to the development of drugs with improved palatability, and research from this perspective is expected to continue.

Although the mean gLMS value of the prednisolone powder, which most closely mimics the formulation actually prescribed to children in Japan, was relatively lower than those of the prednisolone-saturated solution and 1 mM quinine, the gLMS ratings of prednisolone powder were “strong” (34.7) or higher for 22 subjects (Figure 2) and “very strong” (52.5) or higher for 12 subjects (Figure 1). Prednisolone has very low solubility in water [12]. Therefore, when it is placed in the oral cavity for a short duration as performed in this study, only a very small amount is likely to dissolve and, thus, rinsing with water is expected to eliminate most of the bitterness. However, our results show that prednisolone powder had a considerable bitter taste. This suggests that despite its low solubility, sufficient amounts of prednisolone are dissolved in the oral cavity in a short time, thereby producing a bitter taste. Alternatively, since prednisolone is highly fat-soluble, it may have adsorbed onto the cells in the oral cavity, resulting in the perception of bitterness. Nonetheless, the actual explanation is unknown. Although individual differences in sensitivity and differences in dosage by target indications or body weight should be considered, the fact that the pharmaceutical preparation containing only 1% prednisolone produced a strong bitter taste in adults implies the extent of bitterness experienced by pediatric patients. Notably, this study was conducted in adults; therefore, age differences should also be considered when interpreting the study results, as age may be a factor influencing the perception of prednisolone bitterness.

In summary, although certain individual differences exist, the results of the single-blind sensory test in adults revealed that prednisolone has a strong bitter taste comparable to the taste of quinine. As we were not able to identify the cause of individual differences in the perception of bitterness in this study, it remains important to consider the bitterness of drugs in clinical settings until more determinants of bitterness are elucidated. For example, systems for evaluating bitterness and identifying individual differences in taste perception can be established. For good medication adherence, it is important to develop pharmaceutical formulations in which bitterness is masked or simply select a less bitter drug if there are multiple drugs with the same indication and varying bitterness levels. As the precise mechanisms underlying drug bitterness perception and the reasons for the individual differences in the perception become clearer, the development of drug substances or products that do not taste bitter to the majority of the recipients will become easier. The realization of better-tasting drugs may lead to improved medication adherence, which will, in turn, enhance the therapeutic effects to optimal levels.

## Figures and Tables

**Figure 1 pharmaceutics-14-02454-f001:**
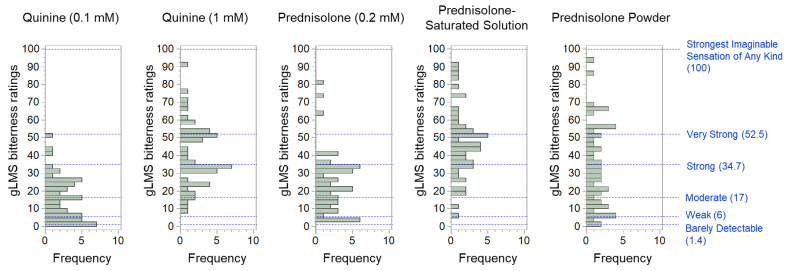
Histograms of the gLMS bitterness ratings by subjects (*n* = 50) for each stimulus.

**Figure 2 pharmaceutics-14-02454-f002:**
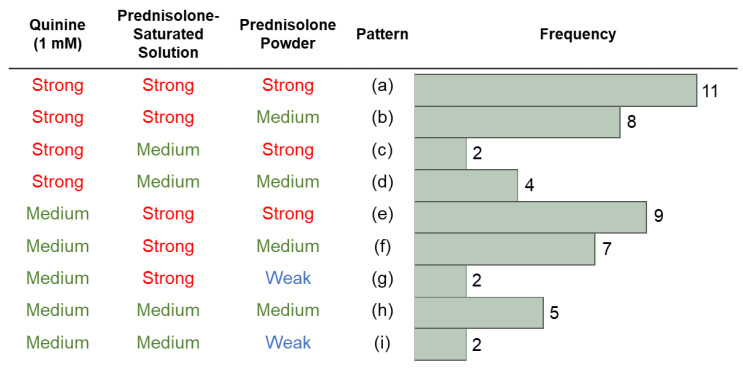
Patterns of subjects’ bitterness perception.

**Table 1 pharmaceutics-14-02454-t001:** Five stimuli were used when conducting the sensory test.

Name of Stimuli	Weight and Name of Active Constituent	Solvent/Dispersion Medium	Total Volume/Weight
Quinine solution (0.1 mM)	0.40 mg quinine hydrochloride	Water	10 mL
Quinine solution (1 mM)	4.0 mg quinine hydrochloride	Water	10 mL
Prednisolone solution (0.2 mM)	0.72 mg prednisolone	Water	10 mL
Prednisolone-saturated solution	Approximately 2.93 mg prednisolone	Water	10 mL
Prednisolone powder (1%)	2.0 mg prednisolone	Lactose	0.2 g

**Table 2 pharmaceutics-14-02454-t002:** The gLMS bitterness rating scores by subjects (*n* = 50) for the bitter taste of each stimulus.

	Quinine (0.1 mM)	Quinine (1 mM)	Prednisolone (0.2 mM)	Prednisolone-Saturated Solution	Prednisolone Powder
Mean	15.9	40.1	24.5	46.8	34.1
Min.	0.7	8.7	2.7	6.0	2.0
Max.	50.7	90.0	80.0	92.0	94.7
SD	12.2	18.0	16.8	19.7	23.0
CV	0.77	0.45	0.69	0.42	0.67

**Table 3 pharmaceutics-14-02454-t003:** Genotypes for *TAS2R19* in subjects rated 1 mM quinine as equal to or more than “strong” or below “strong” (medium or weak). The variants G and A result in arginine and cysteine respectively of the TAS2R19 299th amino acid (through the complementary strand). The *p*-value was calculated by Fisher’s exact test.

*TAS2R19* Genotype	Strong	Medium or Weak	*p*-Value
GG	11	15	0.0266
AG	14	6	
AA	0	3	
	25	24	

## Data Availability

All relevant data are included in the article or Appendix A.

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
