# Peer review of "Comparison of Bitterness Intensity between Prednisolone and Quinine in a Human Sensory Test Indicated Individual Differences in Bitter-Taste Perception"

_pharmaceutics, 2022, doi:10.3390/pharmaceutics14112454_

Round 1

Reviewer 1 Report

The manuscript entitled "Comparison of Bitterness Intensity Between Prednisolone and Quinine in a Human Sensory Test Indicated Individual 3 Differences in Bitter-Taste Perception" represents an interesting research focused on investigation of a steroid drug bitterness in comparison to a standard. In this regard, the novelty of the manuscript is questionalble since the authors referred to two papers dedicated to prednisolone bitterness. so this property of the drug is not new. In this sense, the main goal to confirm bitterness of the drug seems pointless. It would benefit the work if the authors suggested an approach to mask the bitter taste and compare to the standard. In fact, the significantce of the paper is in its second part dedicate to investigating the relationship between the drug bitterness and genetic polymorfsm. However, I believe this topic would suit another journal.

Still, the paper is well structured, literally written, methods are clerly described and the results are properly visualized. Furthermore, satisfactory discussion was presented and supported by adequate conclusions.

I would recommend revision of the title in order to have a shorter and clear version.

Author Response

Point 1: I would recommend revision of the title in order to have a shorter and clear version.

Response 1: Thank you for your comment. The authors believe that the significance of this study is that in addition to the perception of the bitterness of prednisolone, there were individual differences in the perception of the bitterness of quinine, the standard substance for bitterness, and it differed by subjects that either or both substances were perceived as strongly bitter. Therefore, the point of this study was to compare the bitterness of quinine with the bitterness of quinine, which is the proposed title of our submission. If it is still too long and unclear, we will consider revising it again.

Author Response

Point 1- First of all the manuscript should be checked by an English native speaker to remove the syntax and typos.

Response 1- The submitted manuscript has been natively checked by Editing & Proofreading Services, Editage (https://www.editage.jp/). We have attached editing certificate issued by Editage this time.

Point 2- The abstract should be modified to give more digital results rather than elastic sentences.

Response 2- In light of your comment, we described the actual score of gLMS in the abstract (line 26).

Point 3- The introduction should provide more details about the previous work

Response 3- Considering your comment, we have described the previous studies cited (references 5 and 6) in the text of the introduction in detail (page 2, lines 54-58).

Point 4- The authors should clarify the meaning of 1% prednisolone powder

Point 5- Please clarify the following sentence: “Five stimuli (0.1 mM quinine, 1 mM quinine, 0.2 mM prednisolone, prednisolone saturated solution, 1% prednisolone powder, and 10 mL as a solution or 0.2 g as a powder) were used for the sensory test. These stimuli contained 0.40 mg or 4.0 mg of quinine hydrochloride, or 0.72 mg, 2.93 mg (approximate), or 2.0 mg prednisolone, respectively”

Response 4 and 5- We believe that the explanation of stimuli used in this study was difficult to understand from the previous manuscript, so we have made a list in the text to make it easier to understand (page 3, lines 115-124).

Point 6- The standard deviation bars should be added to the figures

Response 6- Thank you for your comment. There is no standard deviation bar that can be shown in the figures, because Figure 1 is a histogram showing the number of participants in each gLMS score range, and Figure 2 shows the number of participants included in each pattern as a bar graph. In relation to Figure 1, instead we describe the standard deviation for gLMS values ​​of each stimulus in Table 1. We would appreciate your confirmation of whether there are any problems with these descriptions.

Point 7- The references need updating to include 2021-2022

Response 7- Thank you for pointing this out. We have tried to include the most recent findings as references. Although many studies concerning taste reception of standard substances or food exist, there are not many studies on the mechanisms of drug bitterness perception, so we had no choice but to cite some older papers. We believe that we have included a large number of relatively new references. For your information, we have added one additional citation in this resubmission, bringing the total number of references to 21.

Point 8- The authors should state whether the participant had been trained to the gLMS before being enrolled in the study or not

Response 8- Thank you for pointing this out. The participants had not been trained with the gLMS before being enrolled in the study. We have added the description to page 4, lines 149-150 of the resubmitted manuscript.

Point 9- The authors should use symbols as *, + on all figures to show the statistical differences

Response 9- Thank you for your comment. As it is difficult to describe the statistical significances between the bitterness intensity scores (gLMS) for each drug in the presented figures, we have described the result in the text along with the p-value (pages 4-5, lines 194-197).

Reviewer 3 Report

It is a well designed, undertaken and reported sensory study. I have only minor comments to make:

1) part 2.2 is not clear despite being quite important to the nature of the work. The authors should really find a better way to describe their samples (liquid -solution or suspension?  or solid and conc in MM and mg/ml) maybe a table would ease?

2) Ethnicity is completely omitted in the discussion but age/sex and lifestyle is mentioned - I guess the participants were all Japanese? how does that compare to ref 13?

3) why after taste is not presented to support the theory of adsoption of prednisolone on cells in the oral cavity?

4) the conclusion is not supported by findings and is not clear (the last 8 lines).

Author Response

Point 1: Part 2.2 is not clear despite being quite important to the nature of the work. The authors should really find a better way to describe their samples

(liquid -solution or suspension? or solid and conc in MM and mg/ml) maybe a table would ease?

Response 1: We have made explanations of the stimuli used in this study to a list to make it easier to understand (page 3, lines 115-124).

Point 2: Ethnicity is completely omitted in the discussion but age/sex and lifestyle is mentioned - I guess the participants were all Japanese? how does that compare to ref 13?

Response 2: All participants in this study were Japanese. As you pointed out, previous studies suggest that there are differences in the perception of taste by race. In addition to differences in genetic backgrounds such as polymorphism of taste receptor genes, it is believed that differences in dietary habits also affect the perception. Therefore, there may be differences in the perception of prednisolone among ethnic groups. We would like to examine this point in the future.

Point 3: Why after taste is not presented to support the theory of adsoption of prednisolone on cells in the oral cavity?

Response 3: Thank you for your comment. In light of your comment, we have added the aftertaste data as Supplementary Material Table S2. We hypothesized that the strong bitterness of prednisolone, which is a poorly soluble drug, is due to its adsorption to cells in the oral cavity. This hypothesis is seemed to be related to aftertaste as you mentioned. However, since no significant difference in aftertaste was observed among the various drugs under the conditions of this study, further investigation is necessary to test the hypothesis. As the next step, we think it is necessary to introduce a new evaluation system that measures the persistence of bitterness in the oral cavity in more detail and examine the relation between aftertaste and residual rate of drug in the oral cavity.

Point 4: The conclusion is not supported by findings and is not clear (the last 8 lines).

Response 4: Based on your comment, we have revised the descriptions in the last paragraph (lines 320-329) to include the relevance of this study.